# Increase in Efficacy of Checkpoint Inhibition by Cytokine-Induced-Killer Cells as a Combination Immunotherapy for Renal Cancer

**DOI:** 10.3390/ijms21093078

**Published:** 2020-04-27

**Authors:** Mojgan Naghizadeh Dehno, Yutao Li, Hans Weiher, Ingo G.H. Schmidt-Wolf

**Affiliations:** 1Department of Integrated Oncology, CIO Bonn, University Hospital Bonn, Venusberg-Campus 1, D 53127 Bonn, Germany; naghizadeh.mojgan@gmail.com (M.N.D.); Yutao.li@ukbonn.de (Y.L.); 2Department of Applied Natural Sciences, Bonn-Rhein-Sieg University of Applied Sciences, D-53359 Rheinbach, Germany; Hans.Weiher@h-brs.de

**Keywords:** cytokine-induced killer cells, monoclonal antibody, PD-1/CTLA-4, immunotherapy, renal cancer

## Abstract

Cytokine-induced killer (CIK) cells are heterogeneous, major histocompatibility complex (MHC)-unrestricted T lymphocytes that have acquired the expression of several natural killer (NK) cell surface markers following the addition of interferon gamma (IFN-γ), OKT3 and interleukin-2 (IL-2). Treatment with CIK cells demonstrates a practical approach in cancer immunotherapy with limited, if any, graft versus host disease (GvHD) toxicity. CIK cells have been proposed and tested in many clinical trials in cancer patients by autologous, allogeneic or haploidentical administration. The possibility of combining them with specific monoclonal antibodies nivolumab and ipilimumab will further expand the possibility of their clinical utilization. Initially, phenotypic analysis was performed to explore CD3, CD4, CD56, PD-1 and CTLA-4 expression on CIK cells and PD-L1/PD-L2 expression on tumor cells. We further treated CIK cells with nivolumab and ipilimumab and measured the cytotoxicity of CIK cells cocultured to renal carcinoma cell lines, A-498 and Caki-2. We observed a significant decrease in viability of renal cell lines after treating with CIK cells (*p* < 0.0001) in comparison to untreated renal cell lines and anti-PD-1 or anti-CTLA-4 treatment had no remarkable effect on the viability of tumor cells. Using CCK-8, Precision Count Beads™ and Cell Trace™ violet proliferation assays, we proved significant increased proliferation of CIK cells in the presence of a combination of anti-PD-1 and anti-CTLA-4 antibodies compared to untreated CIK cells. The IFN-γ secretion increased significantly in the presence of A-498 and combinatorial blockade of PD-1 and CTLA-4 compared to nivolumab or ipilimumab monotreatment (*p* < 0.001). In conclusion, a combination of immune checkpoint inhibition with CIK cells augments cytotoxicity of CIK cells against renal cancer cells.

## 1. Introduction

Cytokine-induced killer (CIK) cells are a heterogeneous cell group consisting of CD3^+^CD56^+^, CD3^-^CD56^+^ and CD3^+^CD56^-^ T cells, which exercise their cytotoxicity in a non-major histocompatibility complex (MHC)-restricted manner [1]. The CD3^+^CD56^+^ subpopulation serves as the main effector cells combining T cell capability with NK cell function. Tumor lysis might exert functionally via the LFA-1/ICAM-1 pathway or NKG2D receptor and MHC-related ligands (MIC A/B) and the ULBP family on tumor cells, which results in up-regulated secretion of perforin and granzyme [2,3,4]. CIK cells have already been proven to be a promising treatment against malignant diseases in various preclinical and clinical studies [5,6,7,8,9]. Previous clinical studies indicated that the adjuvant immunotherapy with CIK cells might prevent recurrence and improve life quality and disease-free survival rates of cancer patients [10]. However, due to immune resistance of cancer cells, the therapeutic activity of adoptive CIK cells is not as efficient as anticipated. The possible explanation for the immune resistance of cancer cells, was mainly mediated by both the immune “checkpoint” programmed death-1 (PD-1) pathway and the negative immune regulation of T cell surface transmembrane cytotoxic T-lymphocyte-associated protein 4 (CTLA-4) receptor signaling pathway [11].

The PD-1 (programmed cell death-1) receptor is expressed on the surface of activated T cells. Its ligands, PD-L1 and PD-L2, are commonly expressed on the surface of dendritic cells or macrophages. PD-L1 is mainly expressed in various tumors, such as cervical, colon, gastric, HBV-related hepatocellular carcinoma (HCC), HCC, melanoma, non-small cell lung cancer (NSCLC), ovarian and renal cell carcinoma (RCC). PD-1 and PD-L1/PD-L2 belong to the family of immune checkpoint proteins that act as coinhibitory factors that can halt or limit the development of the T cell response including activation of signaling pathway of T-cell receptor (TCR) as well as the secretion of immune-stimulatory cytokines [12,13]. CTLA-4 is homologous to the T-cell costimulatory protein, CD28, and both molecules bind to CD80 and CD86, also called B7-1 and B7-2 respectively, on antigen-presenting cells. The mechanism by which CTLA-4 acts in T cells remains controversial. Biochemical evidence suggested that CTLA-4 recruits a phosphatase to the T cell receptor (TCR), thus attenuating the signal [14]. More recent work has suggested that CTLA-4 may function in vivo by capturing and removing B7-1 and B7-2 from the membranes of antigen-presenting cells, thus making these unavailable for triggering of CD28 [15].

The immune checkpoint inhibitors have been shown to be efficacious in patients with advanced renal cell carcinoma. Dual immune checkpoint inhibition PD-1 and CTLA-4 increases CD8^+^ lymphocytes infiltrating renal cancer [16]. The European Association of Urology Guidelines Panel has updated its recommendations since a front-line ipilimumab and nivolumab combination therapy has demonstrated a survival benefit in recent randomized trials for renal carcinoma patients [17]. In clinical trials, overall survival, objective response rates and health-related quality of life (HRQoL) were significantly higher with nivolumab plus ipilimumab than with sunitinib among intermediate and poor risk patients with previously untreated advanced renal cell carcinoma [18,19].

The aim of this project was to investigate the potential synergistic effects of nivolumab and ipilimumab on cytotoxicity of CIK cells in coculture with the renal cell carcinoma (RCC) cell line A-498 and Caki-2. There has been little information about adaptive immunotherapy with immune checkpoint inhibitors in renal cancers. In this study, we provided new evidence that PD-1 and PD-L1 blockade combination strengthen tumoricidal activity of CIK cells on RCC. This suggests that the combination immune checkpoint inhibitors with CIK cells represent a promising approach in the future treatment of patients with renal carcinoma.

## 2. Results

### 2.1. In Vitro Expansion and Phenotypic Characteristic of CIK Cells

After 14 days in vitro expansion, the majority of CIK cells had a CD3^+^CD8a^+^ phenotype with a percentage of 74.7% ± 3.5%. Notably, the subset of NKT cells with coexpression of CD3 and CD56 was 68.7% ± 0.8%, in contrast to a coexpression of CD3 and CD4 of 27.7% ± 0.7% (Figure 1).

### 2.2. Surface Expression of Immune Checkpoint PD-1 and CTLA-4 on CIK Cells and PD-L1/PD-L2 on A-498 or Caki-2 Renal Cell Lines

Flow cytometric analysis was conducted to determine the cell surface expression of immune checkpoint inhibitors PD-1 and CTLA-4 on CIK cells and PD-L1/PD-L2 expression on A-498 or Caki-2 cells. We found that the percentage of CD3^+^PD-1 on surface CIK cells was significantly higher than that of CD3^+^ CTLA-4 CIK cells (3.9% ± 0.5% versus 1.3% ± 0.3%, *p* < 0.001). Additionally, PD-L1 surface expression on Caki-2 was remarkably higher than A-498 (96.5% ± 0.1% versus 94.9% ± 0.9%, *p* = 0.02) while there was no difference on PD-L2 expression (1.4% ± 0.1% versus 1.8% ± 0.1%, *p* = 0.66; Figure 2).

### 2.3. Effects of CIK Cells Against Renal Cell Lines

In this assay, the cytotoxicity of CIK cells against renal cell lines was investigated. After 8 days of CIK cell generation, CIK cells at varying effector/target ratios (20:1, 10:1, 5:1 and 1:1. CIK cells represent effector cells, tumor cells represent target cells) were cocultured with the renal cell lines, A-498 and Caki-2 for 72 h. As controls, untreated renal cell lines were used. CCK-8 assay results showed that at 72 h after treatment with CIK cells, the cell viability significantly decreased in the effector:target (E:T) ratio of the 5:1, 10:1 and 20:1 group of A-498 and Caki-2, respectively (Figure 3A,B).

Figure 3A shows a significant decrease in viability of A-498 at E/T ratio of 10:1 about 50% cells comparing to control. Increasing the E/T ratio from 1:1 to 20:1 led to a significant drop to a viability of 40%. However, there was no significant difference at E/T 1:1 ratio as compared to the control. Figure 3B displays that the viability of tumor cells Caki-2 decreased with an increasing E/T ratio. There was a significant decrease in the viability of Caki-2 at the E/T ratio of 20:1 to about 50% cells comparing to the control. Conversely, there were no significant differences at E/T 1:1 compared to the control. Hence, CIK cells exerted stronger cytotoxicity against A-498 compared to Caki-2.

### 2.4. Effects of Nivolumab and Ipilimumab on Renal Cell Lines Culture

The purpose of this assay was to assess potential effects of nivolumab and ipilimumab on the viability of the A-498 and Caki-2 cells. Renal carcinoma cell lines were treated with 20 μg/mL nivolumab, 20 μg/mL ipilimumab and a combination of both drugs with the same concentration for 72 h and compared to untreated cell lines.

Figure 4 shows the effects of nivolumab and ipilimumab on the viability of renal cells. In Figure 4A there was no statistically significant difference in viability of A-498 after treatments with respective drugs compared to untreated A-498 (*p* = 0.3598). Likewise, there was no remarkable difference on the viability of Caki-2 cells with treatments of immune checkpoint inhibitors (*p* = 0.2658, Figure 4B). Taken together, there was no significant effect of a combination treatment of nivolumab and ipilimumab on the viability of A-498 and Caki-2 cell lines.

### 2.5. Effects of Nivolumab and Ipilimumab on CIK Cell Proliferation

In this assay, possible effects of nivolumab and ipilimumab on the quantity of the CIK cells were investigated. CIK cells from three different healthy donors, at 8th day of the culture, were treated with either 20 μg/mL nivolumab or 20 μg/mL ipilimumab or a combination of both drugs with the same concentration for 72 h in comparison to untreated CIK cells.

To analyze the percentages of the viable CIK cells after antibody treatment, viable cells were determined using a CCK-8 cell proliferation cytotoxicity assay. It revealed that the proliferation significantly increased after a combination treatment compared to untreated CIK cells (*p* = 0.0087). We tested the counts of viable CIK cells after treatments using trypan-blue counting method. In each well, 5 × 10^4^ CIK cells were seeded initially. After 3 days of incubation with antibodies, there was a statistically significant increase in the number of CIK cells after treatment with a combination of nivolumab and ipilimumab in comparison to untreated CIK cells (*p* = 0.0227). We further counted the absolute number of cells utilizing flow cytometry and found that the absolute number of live cells significantly increased after treatment with either a combination of antibodies or ipilimumab compared to untreated CIK cells after 48 h (119.96 ± 4.96 cells/μL versus 82.52 ± 5.60 cells/μL, *p* = 0.042; 127.36 ± 5.00 cells/μL versus 82.52 ± 5.60 cells/μL, *p* = 0.016, respectively).

In summary, a combination treatment of nivolumab plus ipilimumab showed no significant decrease of the viability of A-498 and Caki-2 cells. Conversely, there was a statistically significant increase in the number of CIK cells with the combination treatment compared to untreated CIK cells.

Figure 5 presents the proliferation of the CIK cells with nivolumab and ipilimumab after 3 days of incubation. Figure 5A shows the percentages of the viable CIK cells after antibody treatment using a CCK-8 cell proliferation assay. We found that the proliferation of CIK cells significantly increased after all treatments compared to untreated CIK cells (*p* = 0.0087). Figure 5B,C displays the counts of viable CIK cells after treatments, using the trypan-blue manual counting method and automatic Precision Count Beads™ counting. The initial number of CIK cells was 5 × 10^4^. There were statistically significant increases in the number of CIK cells after a combination treatment with nivolumab and ipilimumab compared to the untreated control (*p* = 0.0227 and *p* = 0.042, respectively).

Figure 6 shows the percentages of the viable cells in a coculture of Caki-2 cells and CIK cells system with nivolumab and ipilimumab treatment in comparison to untreated CIK cells. CIK cells cocultured with Caki-2 significantly decreased the viability of Caki-2 by 40% (*p* < 0.0001). Figure 6A displays a slightly lower viability after treatment with nivolumab plus ipilimumab compared to untreated CIK cells, but there was no statistically significance (*p* = 0.3408). However, the counted number of viable CIK cells was significantly enhanced after nivolumab and ipilimumab treatment (*p* = 0.0206) using the trypan-blue counting method compared to untreated CIK cells (Figure 6B).

Figure 7A,B show the percentages of viable cells compared to the untreated CIK cells after treatment with nivolumab and ipilimumab in a coculture of CIK cells with the A-498 cell line system. After coculture with CIK cells, the viability of A-498 cells decreased to 50% (*p* < 0.0001). Figure 7A shows the percentage of cell viability reduction to approximately 30% after treatment with nivolumab plus ipilimumab, but there was no statistically difference compared to untreated CIK cells (*p* = 0.4425). Figure 7B displays the number of viable CIK cells significantly increased when they were cocultured with A-498 with a combination treatment of nivolumab and ipilimumab by trypan-blue staining, comparing to untreated CIK cells (*p* = 0.0210).

Figure 8 presents the proliferation ability of CIK cells after immune checkpoint inhibitors treatments. The histogram shows a significant reduction in intensity of CellTrace™ dye after 3 days CIK cells culture (*p* < 0.0001). The mean intensity decreased more rapidly than in untreated CIK cells after treatment with nivolumab and ipilimumab (Table 1).

### 2.6. Effects of Nivolumab and Ipilimumab on Proliferation of CIK Cells Coculture with Renal Cell Lines by Flow Cytometry

We further investigated the possible effects of nivolumab and ipilimumab on the viability of CIK cells and A-498 and Caki-2 cell lines using CellTrace ™ violet. Permanently labeled cells with Invitrogen CellTrace™ fluorescent stains without affecting morphology or physiology to trace generations or divisions in vivo or in vitro. Through subsequent cell divisions, daughter cells receive approximately half of the fluorescent label of their parent cells, allowing the analysis of the fluorescence intensities of cells labeled and grown in vivo. Analysis of the level of fluorescence in the cell populations by flow cytometry permits the determination of the number of generations through which a cell be progressed since the label was applied. Prior to coculture CIK cells and renal carcinoma cells, CIK cells on day 8 were stained with CellTrace™ Violet dye, then CIK cells were treated with either 20 μg/mL nivolumab or 20 μg/mL ipilimumab or a combination of both inhibitors for 4 h and then cocultured with renal cells for 72 h.

We analyzed the proliferation of labeled CIK cell after 72 h of coculture to A-498 and Caki-2 on histograms. The intensity means of CellTrace™ violet significantly reduced after coculture with A-498 compared to CIK cells monoculture (*p* < 0.0001, Figure 9A). Moreover, there was a significant decrease in the mean intensity of CellTrace ™ violet after treating with nivolumab plus ipilimumab compared to untreated CIK cells (Table 1). Likewise, the mean intensity of the labeled CIK cells significantly reduced in the presence of Caki-2 comparing to untreated CIK cells (*p* < 0.0001, Figure 9B). In addition, there was a significant decrease in the mean intensity of CellTrace violet after treating with both drugs compared to untreated CIK cells (Table 1). Collectively, the proliferation of CIK cells in the presence of tumor cells increased in contrast to untreated CIK cells.

Therefore, the proliferation of CIK cells remarkably enhanced in the presence of A-498 or Caki-2 after treatment with nivolumab plus ipilimumab compared to monotherapy of nivolumab or ipilimumab. The flow cytometric results showed that the proliferation rate of CIK cells with A-498 cells was significantly higher than that of Caki-2.

Figure 10 demonstrates the effect of a combination of nivolumab and ipilimumab on the proliferation of labeled renal cells cocultured with CIK cells. Figure 10A shows untreated A-498 cells at day 1. Unexpectedly, there were no viable A-498 cells after 72 h coculture with CIK cells. Figure 10B Caki-2 presented the same evidence as A-498. Together, the data indicate that CIK cells exerted the intense cytotoxicity against tumor cells at a ratio of 10:1.

### 2.7. Effects of Nivolumab and Ipilimumab on IFN-Gamma Levels Secreted by CIK Cells

We next investigated the effects of nivolumab and ipilimumab on the secretion of IFN-γ derived from CIK cells as an indicator of their cytotoxic activity when grown in coculture with renal cell lines, A-498 or Caki-2.

Figure 11 shows the IFN-γ concentrations that were found in the supernatant of CIK cells in coculture with A-498 or Caki-2 cells. Obviously neither A-498 nor Caki-2 secreted IFN-γ. In Figure 11A, the concentration of IFN-γ after 48 h of coculture with A-498 is shown. The IFN-γ secretion increased significantly in the presence of A-498 with treatment of nivolumab and ipilimumab compared to nivolumab or ipilimumab monotreatment (392.8 ± 138.1 pg/mL versus 61.0 ± 14.6 pg/mL, *p* < 0.001; 392.8 ± 138.1 pg/mL versus 155.5 ± 78.6 pg/mL, *p* < 0.001).

Figure 11B shows the data for CIK cells in a coculture Caki-2 system after 48 h with the PD-1 blockade/CTLA-4 blockade treatment or a combination treatment. In contrast to untreated CIK cells, the IFN-γ secretion increased significantly in the presence of Caki-2 cells (68.5 ± 49.0 pg/mL versus 26.9 ± 15.7 pg/mL, *p* = 0.003). However, there was no significant difference on the concentration of IFN-γ after treatment of nivolumab and ipilimumab compared to that of nivolumab or ipilimumab monotreatment. Hence, the IFN-γ secretion by CIK cells increased markedly after treatment with nivolumab and ipilimumab coculture with A-498 cells compared to the monotreatment of nivolumab or ipilimumab.

## 3. Discussion

This study contributes to assessing the immunotherapy agent-mediated antitumor effect of CIK cells on RCC cells. The combination of anti-PD-1 and anti-CTLA-4 antibodies provides synergistic antitumor effects of CIK on RCC cells. Interestingly, the cotreatment with anti-PD-1 and anti-CTLA-4 antibodies induced CIK cells proliferation. In addition, the combined treatment up-regulated the secretion of immune-stimulatory cytokines IFN-γ in CIK cells. These results presented us with the preclinical model of CIK cells combined nivolumab and ipilimumab as a promising strategy for adoptive immunotherapy.

Recently, many clinical trials have confirmed the antibody against the above receptor can effectively suppress immune escape of tumor cells. Evidence from the studies that blockade the PD-1 pathway can enhance antitumor T cell reactivity and promote tumor immune control ignited efforts to develop ICB therapies targeting PD-1 in cancer patients. In 2014 the FDA approved a fully human IgG4κ monoclonal antibody, nivolumab (Opdivo^®^), as the first PD-1-targeting ICB therapy for cancer. This approval was based on the outcome of the CheckMate-037 trial, which revealed improved objective response rates to nivolumab versus investigator’s choice chemotherapy in patients with unresectable or metastatic melanoma whose cancers had progressed following treatment with ipilimumab ± a BRAF inhibitor [20].

In addition to its efficacy in the treatment of melanoma, nivolumab has been shown to exhibit a therapeutic benefit against a wider spectrum of cancers than anti-CTLA-4 immune checkpoint inhibitors. Phase III trials for advanced squamous-cell lung cancer (SCLC) and non-small cell lung cancer (NSCLC), advanced renal cell carcinoma (RCC) and recurrent squamous-cell carcinoma of the head and neck (SCCHN) have all demonstrated survival benefits of treatment with nivolumab over traditional therapies [21,22,23,24]. The benefits of ipilimumab + nivolumab combination therapy have also been reported in advanced RCC patients [18].

In 2011, Ingo Schmidt-Wolf et al. established a registry of clinical trials with CIK cells (www.cik-info.org) and collected 11 studies that include a variety of cancers in their registry. Of the 384 patients where a response was reported, 24 patients had a complete response, 27 patients had a partial response, and 40 patients had a minor response. No severe toxicity was observed after infusion of CIK cells [10]. In another random clinical trial, the 3-year disease free survival (DFS) of operable RCC patients treated with autologous tumor lysate-pulsed dendritic cells cocultured with cytokine induced killer (Ag-DC-CIK) was 96.7% compared with 57.7% in the control group [25]. During the expansion of CIK cells, the total number of CD56^+^ cells increase more than 1000-fold during the generation of CIK cells, by sequentially adding IFN- γ, OKT-3 and IL-2. The higher lytic activity of CIK cells is mainly due to the higher proliferation of CD3^+^CD56^+^ cells and cytotoxic effects of CIK cells against tumor cells is blocked by antibodies of LFA-1 and its counter receptor (ICAM-1) [2]. In the recent study, we revealed a combination of anti-PD-1 and anti-CTLA-4 antibodies directly enhanced the proliferation of CIK cells. Although the purpose of this study was not to show the mechanism of cytotoxicity of CIK cells, but to, however, investigate immune molecular interaction involved in cytotoxicity of CIK that is still valuable for personalization treatment for the patients with renal carcinoma.

In our study, the percentage of CD3^+^CD56^+^CIK cells over 50% demonstrated CIK cells having a cytotoxic function (Figure 1). Although both PD-1 and CTLA-4 expression presented a low percentage, which was 3.9% ± 0.5% and 1.3% ± 0.3%, PD-L1 expression on either A-498 or Caki-2 cells was over 95% (Figure 2). More importantly, INF- γ secretion from CIK cells cocultured with A-498 after a combination of PD-1 and CTLA-4 blockade significantly increased more than monotherapy of the PD-1 blockade or CTLA-4 blockade (Figure 11). Recently, PD-L1 as a predictive biomarker of response for anti-PD-1 and anti-PD-L1 antibodies was controversially discussed depending on the interactions between immune responses, the tumor microenvironment and the dynamic nature of the immune system. Seeber et al. [26] found that PD--L1 expression on RCC tissues was negative respective of the nivolumab therapeutic response. There are no differences on tumor-infiltrating immune cells between patients with higher or lower PD-L1 expression. Hence, in our study, although our results showed that PD-L1 surface expression on Caki-2 was higher than on A-498 cells, it might be not related closely to the CIK cells responses to immune checkpoint inhibitors treatments.

Next, we investigated whether A-498 and Caki-2 have different responses to CIK cells in a different E:T (effector:target) ratio. Obviously, there were significant viability reductions at E:T 5:1, 10:1 and 20:1 when A-498 cells or Caki-2 cells interacted with CIK cells (Figure 3). Furthermore, at E:T 10:1 ratio, the percentage of cell viability approximately dropped by 50% in a coculture A-498 with CIK cells system while it reduced by 30% in a Caki-2 plus CIK cells system. These results suggested that A-498 cells might have higher responses to CIK cells than Caki-2 cells.

We further used CCK-8 assay, trypan-blue and Precision Count Beads™ to detect the efficacy of immune check inhibitors on the CIK cells proliferation. We found that there was no effect of immune check inhibitors on the proliferation of A-498 and Caki-2 (Figure 4). In contrast, there was remarkable impact of either nivolumab or ipilimumab or a combination treatment of nivolumab and ipilimumab on the proliferation of CIK cells using CCK-8 assay, manual trypan-blue staining counting by hemocytometer and automatic counting using flow cytometer (Figure 5). Although there was no statistical effect of PD-1/CTLA-4 blockade on the cell viability in the presence of Caki-2 and CIK cells (Figure 6A) or A-498 (Figure 7A) in comparison to untreated CIK cells, the number of CIK cells demonstrated significantly increased after 72 h of coculture of Caki-2 (Figure 6B) and A-498 (Figure 7B) with an immune check inhibitors treatment.

To overcome the CCK-8 assay limitation where the percentage of cell viability reflected both the proliferation of CIK cells and tumor cells (Figure 6A and Figure 7A), we labeled the CIK cells with CellTrace^TM^ violet and conducted flow cytometry experiments to observe the effect of anti-PD-1 and anti-CTLA-4 antibodies on the proliferation of CIK cells in the absence and presence of tumor cells. Our major finding was the observed increase in CIK cell numbers after a combination treatment of nivolumab and ipilimumab in comparison to untreated CIK cells (Table 1, Figure 8). In the presence of A-498 and Caki-2, either nivolumab or ipilimumab or combinatorial nivolumab and ipilimumab affect the proliferation of CIK cells in comparison to untreated CIK cells. We further investigated that CIK cells coculture with A-498 showed us stronger proliferation ability in response to nivolumab compared to ipilimumab or nivolumab plus ipilimumab, with the percentage of mean intensity reducing by 16.2%, 15.08% and 15.27%, respectively. However, with Caki-2 engagement, the percentage of mean fluorescence intensity of CIK cells reduced by 9.24% with nivolumab, 12.97% with ipilimumab and 12.67% with a combination treatment. Taken together, in a coculture system of CIK cells and A-498, the proliferation of CIK cells with either nivolumab or ipilimumab or a combination is stronger in comparison to that of Caki-2. We further investigated IFN-gamma levels from CIK cells by the ELISA method. There was a dramatic elevation on IFN-gamma expression after treatment with nivolumab plus ipilimumab compared to nivolumab or ipilimumab when CIK cells and A-498 were cocultured together whereas CIK cells had no significant response to antibodies cocultured with Caki-2 cells (Figure 11). One possibility is that A-498 is of papillary renal carcinoma [27], which characterizes the amplification of chromosome 8q, the region where the c-Myc gene resides, contributing to c-Myc overexpression and subsequent pathway activation in tumors [28]. Conversely, Caki-2 is a classic clear-cell renal carcinoma, which expresses wild pVHL [27]. In renal cancer, the development of angiogenesis is in part due to the activation of the phosphatidylinositol-3-kinase (PI3K)/AKT/mechanistic target of the rapamycin (mTOR) pathway. Blockade of immunosuppressive signaling using monoclonal antibodies (mAbs) against PD-L1 or PD-1 induces the activation, differentiation and/or proliferation of tumor-infiltrating T cells via derepression of the PI3K-AKT signaling cascades while Myc is post-translationally stabilized via PI3K-AKT and RAS-MAPK signaling activation [29]. Therefore, A-498 might be more sensitive than Caki-2 in response to PD-1/PD-L1 blockade in vitro. Further work is required to elucidate the exactly different mechanism between A-498 and Caki-2 in PD-1/CTLA-4 pathway under the tumor microenvironment.

In summary, this study showed for the first time that the combination of anti-PD-1 and anti-CTLA-4 antibodies provides synergistic antitumor effects of CIK cells in RCC cells and promotes CIK cell proliferation. Combined therapy of simultaneous CIK infusion and PD-1 plus the CTLA-4 signaling blockade might be put forward as a novel promising strategy for clinical trial especially in the setting of unresectable advanced renal carcinoma patients.

Cytokine-induced killer (CIK) cells represent a realistic approach in cancer immunotherapy in solid tumors. In this study, an ideal therapy strategy of checkpoint inhibitor combination with CIK cells was explored. Overall, in the treatment of renal cancers in vitro, checkpoint inhibition plus CIK cells resulted in an increasing cytotoxicity and a further upregulation of interferon-gamma in the presence of A-498. Therefore, combining inhibitory checkpoint blockade with CIK cells may provide an effective strategy to increase therapeutic efficacy in renal cancer patients.

## 4. Materials and Methods

### 4.1. Nivolumab and Ipilimumab

Nivolumab -BMS-936558.20110808- (Lot#U20140606.TB.00903) with the concentration of 5 mg/ mL and ipilimumab (G1)_XAS_Ab (Lot# U20160608.AB219062.02) with the concentration of 20.7 mg/ mL were both provided by Bristol-Myers Squibb Co. (New York, NY, U.S.A.). Nivolumab and ipilimumab was diluted in fresh prepared PBS to the final concentration of 20 μg/mL. Ultra-LEAF™ purified human IgG4 Isotype control antibody (Biolegend, San Diego, CA, U.S.A.) was used as a control antibody on CIK cells for nivolumab and human IgG1 isotype-control antibody was used as a control antibody for ipilimumab. Both control antibodies were diluted in PBS to the final concentration of 20 μg/mL.

### 4.2. Cell Culture

Two human renal carcinoma cell lines A-498 and Caki-2 were obtained from DSMZ (Braunschweig, Germany) or ATCC (Manassas, VA, U.S.A.). A-498 cell lines were cultured in EMEM medium (ATCC, Manassas, VA, U.S.A.), supplemented with 10% heat inactivated FCS (Gibco^®^ by life technologies™, Carlsbad, CA, U.S.A.) and 1% penicillin/streptomycin (Gibco^®^ by life technologies™, Carlsbad, CA, U.S.A.).

Caki-2 cell lines were cultured in McCoy’s 5A medium (Gibco^®^ by life technologies™, Carlsbad, CA, USA), and supplemented with 10% heat inactivated FCS (Gibco^®^ by life technologies™, Carlsbad, CA, U.S.A.) and 1% penicillin/streptomycin (Gibco^®^ by life technologies™, Carlsbad, CA, U.S.A.). The cell lines were kept at 37 °C in a humidified atmosphere with 5% CO_2_.

For harvesting the confluence cell lines, medium was removed and the flask was rinsed gently with 10 mL of prewarmed sterile 1x PBS. Then 5 mL of 0.05% trypsin-EDTA solution (Gibco^®^ by life technologies™, Carlsbad, CA, U.S.A.) was added and the cells were incubated for 5–10 min at 37 °C. Once the cells appeared detached, 10 mL of the prewarmed medium was added to stop trypsinization. The cell suspension split from 1:3 to 1:5 with the fresh warm medium. To analyze the cell surface marker for flow cytometry, tumor cells were digested by Accutase cell detachment solution (PAN-Biotech GmbH, Aidenbach, Germany) for 8 min as a gentle detachment solution.

### 4.3. Buffy Coats

For CIK cells in vitro expansion, buffy coats were provided from healthy volunteers with informed consent from blood donation service of the Institute of Experimental Hematology and Transfusion Medicine (IHT) of the University Clinic of Bonn (UKB), Germany.

### 4.4. CIK Cells Generation and Phenotypic Detection of CIK Cells and Tumor Cells

CIK cells were isolated from PBMCs derived from healthy blood donors and generated according to the established CIK cell generation standard protocol (20). CIK cells were grown in RPMI 1640 medium with 2.5% (*v*/*v*) of HEPES buffer (both from PAN-Biotech GmbH, Aidenbach, Germany), 10% (*v*/*v*) heat inactivated FCS, 1% P/S (both Gibco^®^ by life science technologies™, Carlsbad, CA, U.S.A.) and were incubated aseptically at 37 °C with 5% CO_2_. The flask incubated at 37 °C with 5% CO_2_ for 1–2 h to get rid of monocytes that adhere to the flask. Of RhIFNγ 1000 IU/mL (R&D Systems Inc., Minneapolis, ME, U.S.A.) was initially added to CIK cells for 24 h, followed by 300 IU/mL interleukin-2 (Novartis Pharma AG, Basel, Switzerland), 50 ng/mL anti-human anti-CD3 mAb and 100 IU/mL interleukin-1β (both from eBioscience Inc., San Diego, CA, U.S.A.). Interleukin-2 was replenished every third day of culture. The cells were used for investigation after eight days of culture.

### 4.5. Phenotypic Detection of CIK Cells and Tumor Cells

The phenotype of CIK cells was detected by a multiparameter flow cytometric analysis using monoclonal antibodies: anti-CD3-FITC, anti-CD56-PE, anti-CD4-APC, anti-CD8a-Brilliant Violet 421, anti-PD-1-APC and anti-CTLA-4-APC (obtained from Biolegend, San Diego, CA, U.S.A.) and 7AAD (a viability probe for methods of dead cell exclusion, BD biosciences, San Diego, CA, U.S.A.). To observe the surface expression of PD-L1/PD-L2 on RCC cell lines, we stained with monoclonal antibodies anti-PD-L1-PE and anti-PD-L2-Brilliant Violet 421 (obtained from Biolegend, San Diego, CA, U.S.A.) and 7AAD.

### 4.6. Cell Counting by Trypan Blue

A Neubauer counting chamber (LO—Labor Optik, Lancing, UK) was used to determine the cell concentration. Trypan blue 0.5% (Biochrom GmbH, Berlin, Germany) was diluted 1:10 in 1x PBS prior to staining. The cell suspensions were diluted and stained in 0.05% trypan blue to ratio between 1:2 and 1:10 based on the original concentration of cells. Viable cells were counted in four quadrants using an H 500 microscope (Helmut Hund GmbH, Wetzlar, Germany) and the cell concentration was calculated by the following formula:(1)$$Cells\;per\;\mathrm{mL}=\frac{\mathrm{counted}\;\mathrm{cells}}{4}\times 10^{4}\times dilution\;factor$$

### 4.7. Cell Counting Kit-8 Cytotoxicity Assay

A-498 and Caki-2 cells were harvested and counted. Then 100 μL containing 1 × 10^3^ cells were seeded into a flat-bottom Nunclon™ 96-well plate (Thermo Fisher Scientific Inc., Waltham, MA, U.S.A) and were incubated at 37 °C with 5% CO_2_ for 24 h. The next day CIK cells were counted and incubated initially for up to 4 h with the nivolumab and ipilimumab or isotype controls. Of the pre-treated CIK cell suspension containing 300 IU/mL IL-2 100 µL was seeded to each well.

Cancer cell lines were cocultured with CIK cells for about 72 h at 37 °C with 5% CO_2_. Then the plates were centrifuged at 1200 rpm for 8 min, after that 100 µL of supernatant was gently removed from the wells and stored at −20 °C for further interferon gamma ELISA assays. According to the protocol, 10 μL of CCK-8 working solution (Dojindo Molecular Technologies, Inc., Rockville, MD, U.S.A.) was added per well and placed in a CO_2_ incubator for about 1–4 h at 37 °C. The absorbance was measured at the wavelength of 450 nm by using a FLUOstar OPTIMA micro plate reader (BMG Labtech, Ortenberg, Germany).

### 4.8. Cell Absolute Counting by Precision Count Beads™

In order to observe the proliferation of CIK cells after treatment with nivolumab or ipilimumab, CIK cells were seeded at a density of 5 × 10^4^ cells per well in 48-well plates. 20 μg/μL nivolumab or 20 μg/μL ipilimumab or 20 μg/μL isotype control were added to wells containing 200 μL of culture medium with 300 IU/mL IL-2, and the plates were incubated for 24 h, 48 h and 72 h at 37 °C with 5% CO_2_, respectively. CIK cells were harvested at three different timepoints and washed with cold DPBS twice. The total cell suspension volume was 300 μL in each tube. Precision Count Beads™ (Biolegend, San Diego, CA, U.S.A.) were vigorously vortexed for 30–40 s to ensure complete mixing and break up of aggregates that may occur during storage. Finally, 3 μL of PerCP-7AAD were added to each tube. Cell number was automatically recorded using flow cytometry. Dead cells were stained with membrane impermeant dye 7AAD and excluded in cell counting. Bead count and CIK cell count was determined by gating on beads and cells as depicted in Figure 5C. Cells were gated on CIK cells, based on their Forwards Scatter (FSC) vs. Side Scatter (SSC) profile. The absolute number of cells could be calculated as follows:(2)$$Absolute\;Cell\;Count\;\left(\frac{Cell}{\mu L}\right)=\frac{Cell\;Count}{Precision\;Count\;Bead™\;Count}\times Precision\;Count\;Beads™\;Concentration\;\left(\frac{Beads}{\mu L}\right)$$

### 4.9. Cell Proliferation Assay Using the CellTrace™ Violet Cell Proliferation Kit

Monitoring of the cell proliferation was performed using a CellTrace™ Violet Cell Proliferation Kit (Life Technologies Corporation, Eugene, OR, U.S.A.). The renal carcinoma cells and CIK cells stained with the CellTrace™ Violet, respectively.

A-498 and Caki-2 cells were harvested and counted. Then cell suspension of 1 mL containing 5 × 10^4^ cells was seeded into the each well of a 6-well-plate and incubated at 37 °C with 5% CO_2_ for 24 h. CellTrace™ stock solution prepared followed by the protocol and the working concentration of 5 µM for staining cells. The media of half of wells containing A-498 and Caki-2 cells were discarded and washed two times with prewarmed PBS then 1 mL prewarmed PBS with 5 μM CellTrace™ was added to them and the plates incubated for 20 min in the dark at 37 °C. After incubation, cells were washed twice with the culture media and replaced with fresh prewarmed culture media. Unstained CIK cells on day 8 after treatment with immune checkpoint inhibitors or isotype controls for 4 h were cocultured with stained tumor cells for 72 h at 37 °C with 5% CO_2_. CIK cells on day 11 were harvested.

After 72 h of incubation, we collected the suspension cells and obtained the attached cells in the wells digested by Accutase solution. Then 1 × 10^5^ cells were transferred to FACS tube and washed twice with 2 mL of 1x PBS followed by centrifugation at 1500 rpm for 8 min. The pellets were resuspended in 1x PBS and were fixed by adding 100 μL of BD CellFIX™ (BD Biosciences, Franklin Lakes, NJ, U.S.A.) covered with Parafilm and stored at 4 °C in the dark until analysis.

To investigate whether nivolumab or ipilimumab affect the proliferation of CIK cells, 5 × 10^6^ CIK cells on day 8 were stained with 5 µM the CellTrace™ and incubated for 20 min in the dark at 37 °C. After being stopped and washed, the CIK cells were incubated with the nivolumab and ipilimumab at concentration 20 µg/mL or isotype controls 20 µg/mL for 72 h. CIK cells on day 11 were obtained, then washed with cold DPBS twice. Pellets were re-suspended in 1x PBS and were fixed covered with Parafilm and stored at 4 °C in the dark until analysis.

### 4.10. Flow Cytometry Analysis

The flow cytometry was performed using a FACS Canto™ II FACS machine (BD Biosciences, San Jose, CA, U.S.A.) and with FACSDiva^TM^ software (BD Biosciences, Franklin Lakes, NJ, U.S.A.). The flow cytometry data was analyzed using FlowJo single cell flow cytometry analysis software v10.3 (FlowJo, LLC, Ashland, OR, USA).

### 4.11. Human Interferon-Gamma ELISA

A Human IFN gamma Uncoated ELISA kit (Invitrogen, Carlsbad, CA, U.S.A) was used to detect the concentration of IFN-γ in cell supernatant. Briefly, Corning Stripwell™ 96-well ELISA Microplates were coated with 100 µL capture antibody overnight at 4 °C. After washing three times the plates were blocked with 200 μL 1xELISPOT Diluent (1x) at room temperature for one hour. Washing steps were repeated and 100 µL of samples or rhIFN-γ standard (1000–15.6 pg/mL) diluted in reagent diluent were added in triplicate and incubated overnight at 4 °C for maximum sensitivity. After incubation plates were aspirated and washed for three times. Then 100 µL detection antibody was added to each well and incubated for one hour at room temperature. During incubation Streptavidin-HRP was diluted to 250:1 using reagent diluent. After incubation and washing 100 µL of Streptavidin-HRP working concentration was added to each well and incubated for 30 min in the dark at room temperature. The washing steps were repeated six times after incubation and 100 µL substrate solution tetramethylbenzidine (TMB) was added. The plates were incubated at room temperature for 15 min in dark. After 15 min of incubation 100 µL of stop solution (Thermo Fisher, Waltham, MA, U.S.A.) was added to each well and mixed thoroughly by tapping.

The absorbance of the plates was read at wavelengths of 450 nm and 570 nm using a FLUOstar Omega Plate Reader (BMG Labtech, FLUORstar Optima, Ortenberg, Germany). The data was analyzed using MARS data analysis with Four-Parameter Curve Fit (BMG Labtech, Orthenberg, Germany). For compensation of potential impurities of the assay plate the absorbance at 570 nm measured.

### 4.12. Statistical Analysis

Statistical analysis was performed with GraphPad Prism version 7.02 software (GraphPad software company, San Diego, CA, U.S.A.). Statistical significance was determined using a Student’s *t* test or one-way ANOVA using a confidence level of 95% by a Dunnett’s multiple comparison post-test. Statistical significance is indicated as * = *p* < 0.05, ** *p* = < 0.01, *** *p* = < 0.001 or **** *p* = < 0.0001.

## Figures and Tables

**Figure 1 ijms-21-03078-f001:**
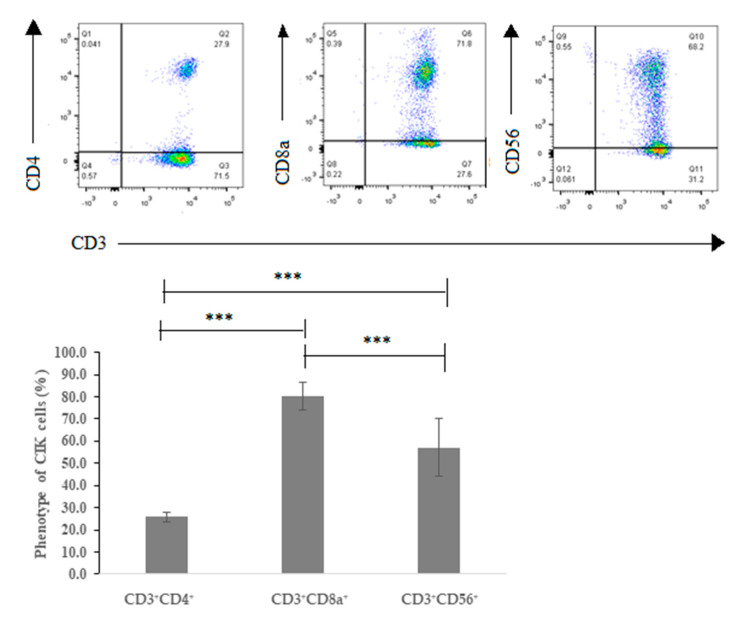
Main phenotype of cytokine-induced killer (CIK) cells derived from donors (*n* = 3) on day 14. Differential expression of three main phenotypic subsets of CIK cells, CD3/CD4/CD8. *** represents a *p* value < 0.001.

**Figure 2 ijms-21-03078-f002:**
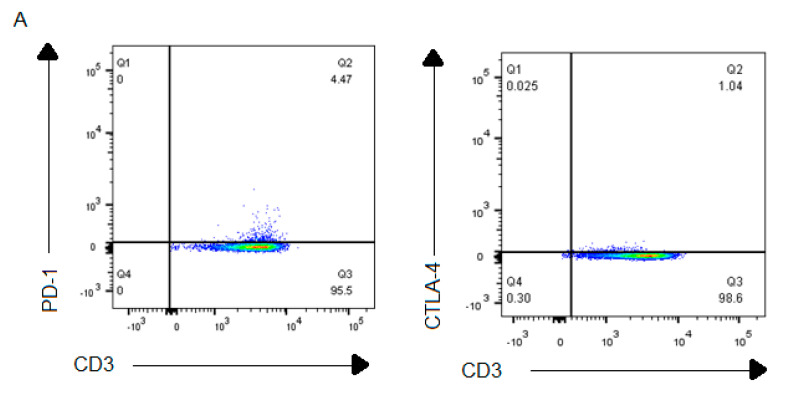
Immune checkpoint inhibitors PD-1/CTLA-4 expression on CIK cells and PD-L1/PD-L2 expression on A-498 and Caki-2 cells. (**A**) Representative flow cytometric bar plots show PD-1 and CTLA-4 expression in CD3^+^ CIK cells. (**B**) Representative flow cytometric histogram plots show the differences in PD-L1/PD-L2 expression on A-498 and Caki-2 cells. The grey filled lines represent the isotype control. The bold lines represent PD-L1/PD L2-stained tumor cells. All the data represents three independent experiments and are shown as mean ± SEM. * represents a *p* value < 0.05, *** represents a *p* value < 0.001.

**Figure 3 ijms-21-03078-f003:**
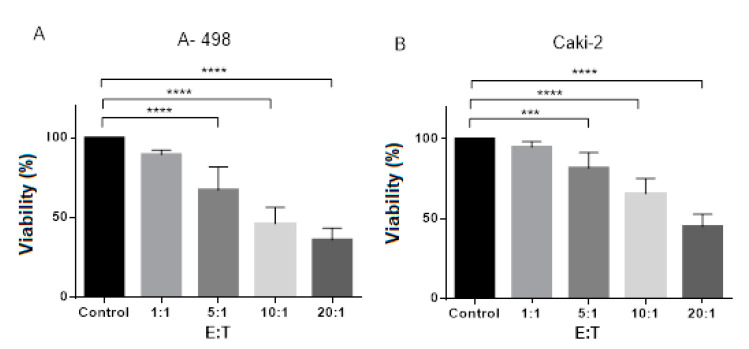
Effects of different CIK cells numbers on the viability of renal cells (effector:target (E:T) ratio) after 72 h of coculture. *n* = 3 healthy donors. (**A**) Coculture of CIK cells and A-498 in different ratios. (**B**) Coculture of CIK cells and Caki-2 in different ratios. Absorbance values have been normalized into percentages with each untreated control showing 100% viability as a reference. *** represents a *p* value < 0.001, **** represents comparing to untreated tumor cells control, a *p* value < 0.0001. E:T ratio represents a ratio of effector cells (CIK cells) and target cells (tumor cells).

**Figure 4 ijms-21-03078-f004:**
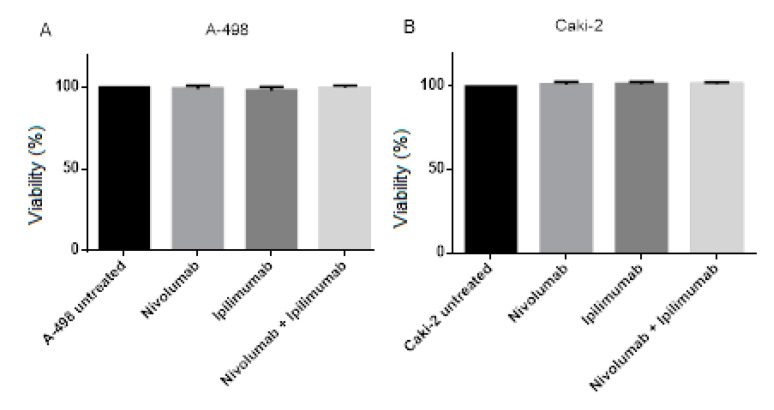
Effects of nivolumab and ipilimumab on the number of viable cells after 72 h on A-498 or Caki-2 cells. (**A**) The different viability of A-498 comparing to A-498 untreated control. *p* value = 0.3598 (**B**) The different viability of Caki-2 comparing to Caki-2 untreated control. *p* value = 0.2658.

**Figure 5 ijms-21-03078-f005:**
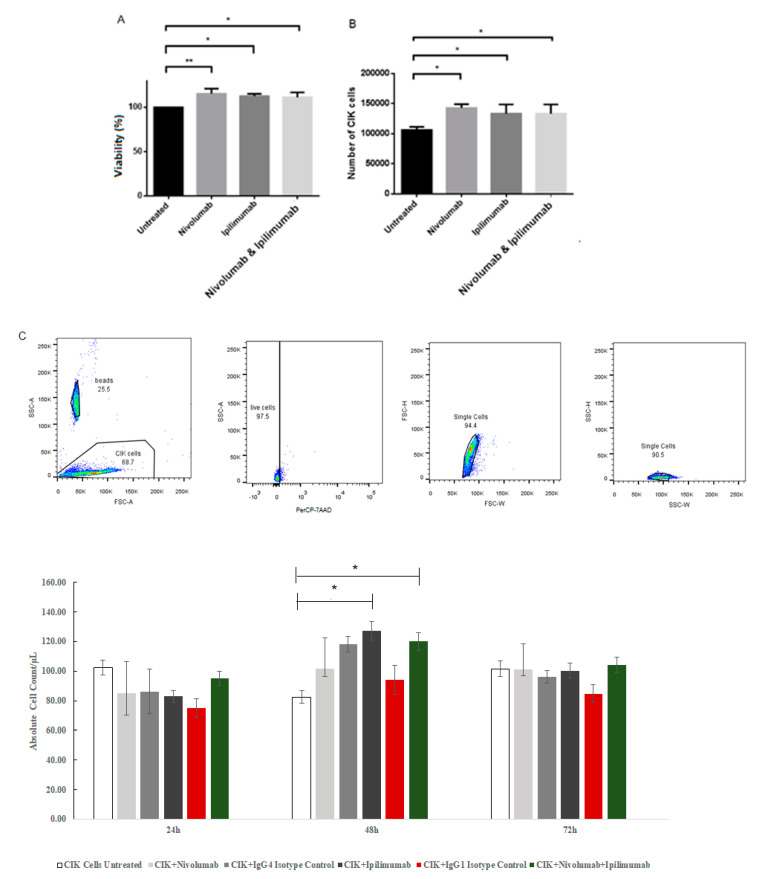
Effects of nivolumab and ipilimumab on the proliferation of CIK cells after 3 days of monoculture using a CCK-8 cell proliferation assay. *n* = 3 healthy donors. (**A**) Percentages of viable CIK cells after a combination treatment with nivolumab and ipilimumab compared to untreated CIK cells using CCK-8 assay, *p* value= 0.0087. (**B**) Number of viable CIK cells after a combination treatment with nivolumab and ipilimumab compared to untreated CIK cells conducted by trypan-blue staining , *p* value = 0.0227. * represents a *p* value < 0.05, ** represents a *p* value < 0.01. (**C**) Overview of the gating strategy and proliferation analysis utilizing Precision Count Beads™. Precision Count Beads™ were added and bead count and live cell count was determined by gating on beads and cells as shown, gating out PerCP-7AAD dead CIK cells. The number of total cells was determined as described in the protocol. * represents a *p* value < 0.05.

**Figure 6 ijms-21-03078-f006:**
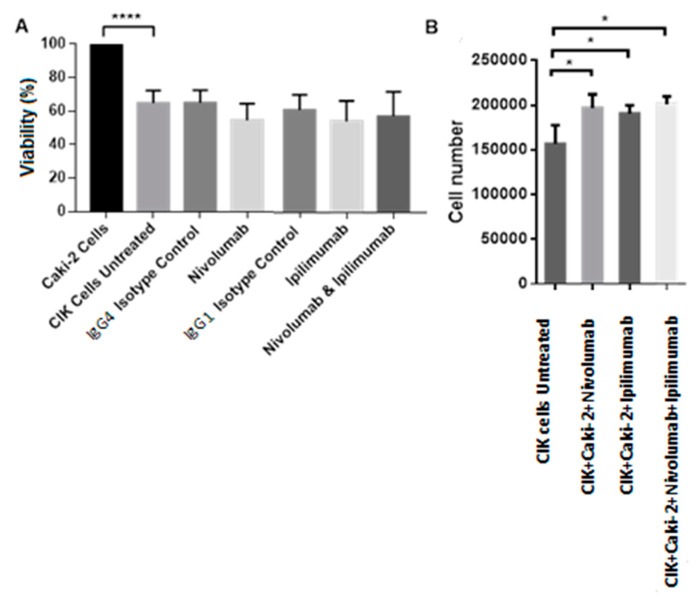
Effects of nivolumab and ipilimumab and control antibodies on the number of viable cells after 72 h of coculture of Caki-2 and CIK cells. *n* = 3 healthy donors. Absorbance values have been normalized into percentages with each untreated control representing 100%. (**A**) Coculture of CIK cells and Caki-2 in a ratio of 10:1 with 20 ug/mL nivolumab. The cell viability after treatment with nivolumab plus ipilimumab compared to untreated CIK cells, *p* = 0.3408. (**B**) Number of viable CIK cells after 72 h treatment with 20 ug/mL nivolumab plus 20 ug/mL ipilimumab in the presence of Caki-2 compared to untreated CIK cells, *p* value= 0.0206. **** represents a *p* value < 0.0001, * represents a *p* value < 0.05.

**Figure 7 ijms-21-03078-f007:**
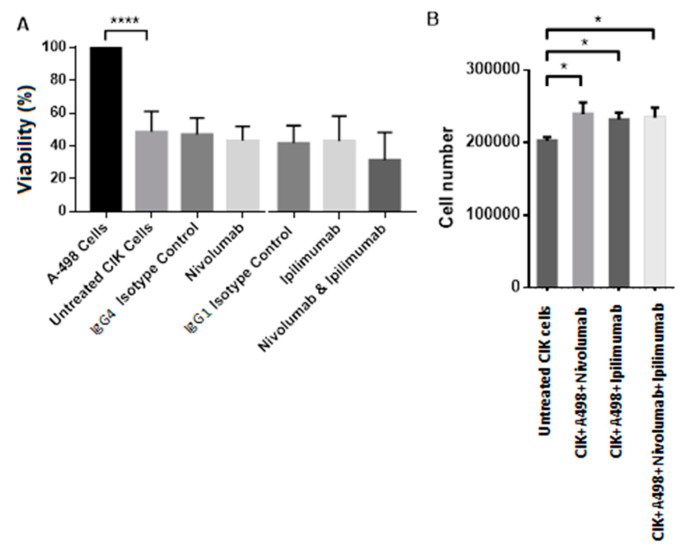
Effects of nivolumab and ipilimumab and control antibodies on the number of viable cells after 72 h of coculture of A-498 with CIK cells. *n* = 3 healthy donors. Absorbance values have been standardized into percentages with each untreated control equaling to 100%. (**A**) Coculture of CIK cells and A-498 in a ratio of 10:1 with nivolumab with a combination treatment of nivolumab plus ipilimumab compared to untreated CIK cells, *p* value = 0.4425. (**B**) Number of viable CIK cells after 72 h treatment with 20 ug/mL nivolumab plus 20 ug/mL ipilimumab in the presence of A-498 compared to untreated CIK cells, *p* value= 0.0210. **** represents a *p* value < 0.0001, * represents a *p* value < 0.05.

**Figure 8 ijms-21-03078-f008:**
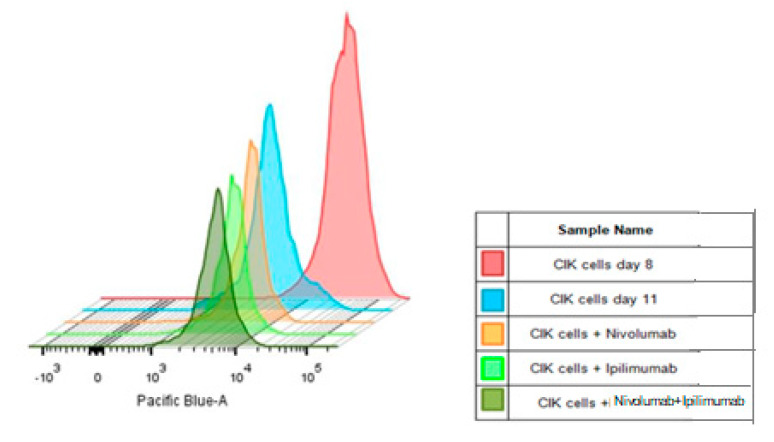
Effects of nivolumab and ipilimumab on proliferation of CIK cells labeled with CellTrace^TM^ violet after 72 h treatment. CIK cells on day 8 were seeded into plates and were administered antibodies. After 72 h incubation, CIK cells on day 11 were harvested and analyzed by flow cytometry. The decrease of fluorescence in labeled CIK cells demonstrated the proliferation of CIK cells.

**Figure 9 ijms-21-03078-f009:**
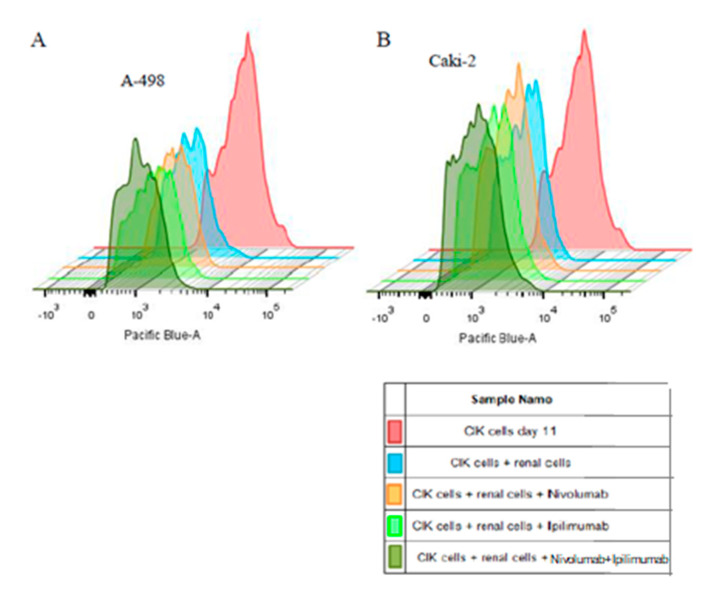
Effects of nivolumab and ipilimumab on proliferation of CIK cells labeled with CellTrace™ violet after 72 h of coculture with renal cells. CIK cells on day 8 were stained with CellTrace™ violet and were administered antibodies for 4 h. Pretreated CIK cells then were cocultured with A-498 or Caki-2 cells. After 72 h of incubation, all cells were harvested and analyzed by flow cytometry. (**A**) Histogram of CIK cells with A-498 coculture. (**B**) Histogram of coculture of CIK cells with Caki-2.

**Figure 10 ijms-21-03078-f010:**
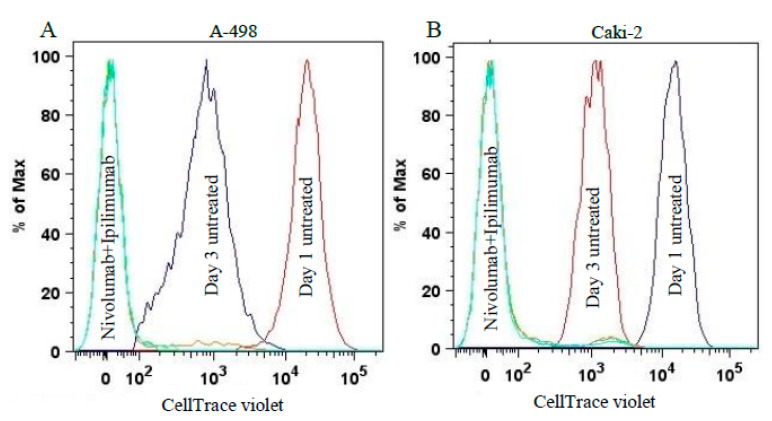
Proliferation of renal cells labeled with CellTrace violet after 72 h of a monoculture and coculture with CIK cells treated with nivolumab and ipilimumab. (**A**) Histogram of the A-498 mono- and coculture with CIK cells. (**B**) Histogram of the Caki-2 mono- and coculture with CIK cells.

**Figure 11 ijms-21-03078-f011:**
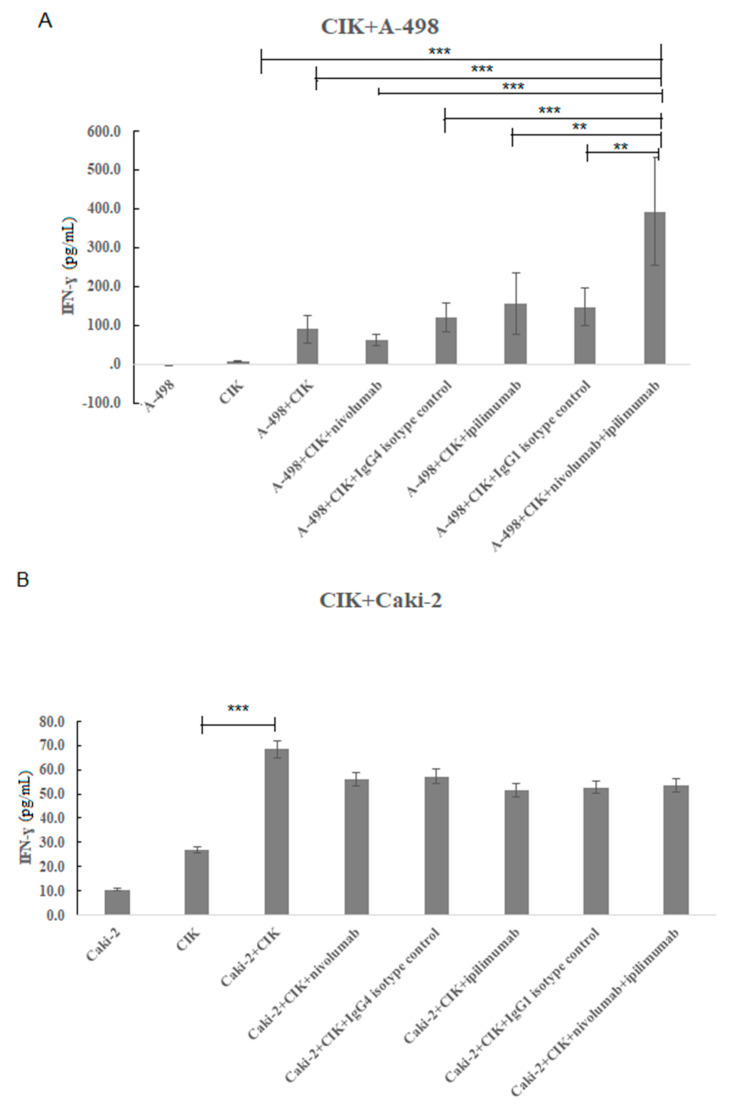
Effects of nivolumab and ipilimumab on IFN-γ levels after 48 h of coculture of CIK cells with A-498 or Caki-2 cell lines. *n* = 3 healthy donors. (**A**) Coculture of CIK cells with A-498 in E/T ratio of 10:1. The IFN-γ secretion increased significantly in the presence of A-498 with treatment of nivolumab and ipilimumab compared to nivolumab or ipilimumab monotreatment, *p* < 0.001 (**B**) Coculture of CIK cells with Caki-2 in a ratio of 10:1. The IFN-γ secretion increased significantly in the presence of Caki-2 cells compared to untreated CIK cells. *p* value = 0.003. *** represents a *p* value < 0.001, ** represents a *p* value < 0.01.

**Table 1 ijms-21-03078-t001:** Mean intensity of CellTrace^TM^ violet of major CIK cells populations measured by flow cytometry analysis after 72 h of the CIK cells co- and monoculture with A-498 and Caki-2. Representative data from one donor. We analyzed the proliferation of labeled renal cells and effects of CIK cells and respective treatments on their proliferation and viability.

Treatment	CellTrace Violet Intensity Mean CIK+A-498	CellTrace Violet Intensity Mean CIK+Caki-2	CellTrace Violet Intensity Mean CIK Cells Alone
Pre incubation	31,000	31,000	31,000
3th day of incubation	1611	1688	7059
Treatment with nivolumab	1350	1532	5953
Treatment with ipilimumab	1368	1469	6091
Combination of nivolumab and ipilimumab	1365	1474	6183

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
