# Peer review of "Increase in Efficacy of Checkpoint Inhibition by Cytokine-Induced-Killer Cells as a Combination Immunotherapy for Renal Cancer"

_ijms, 2020, doi:10.3390/ijms21093078_

Round 1

Reviewer 1 Report

Overall, the study has interesting design and high translational value, however, the description of Methods and Results has to be thoroughly revised both for the correct presentation of procedures and data, as well as for English language.

Major revision:

  1. What was the reason to investigate the effect of nivolumab and ipilimumab on the proliferation of CIK cells for 11 days in monoculture (Fig. 4) while the treatments of cancer cells and co-cultures lasted 72 hours? What is the difference between the results presented in Fig. 4, Fig. 5 C and Fig. 6 C? If panels C for Fig. 5 and Fig. 6 are completely the same, why they are shown in different figures?
  2. In panels A and B of Figures 5, there are the same data for the first two and the last bars. The same is in Fig. 6. It would be reasonable to combine those two figures to one and to avoid the repeated data.
  3. The methods section seems overloaded with details yet unclear and has to be thoroughly revised for English language. Some details brings in confusion. For example, why there is a range of dilutions used for trypan blue staining? Or what does it mean the statement “according to the manuscript” in line 160, in the description of CCK-8 assay? The section 4.7. is named “Cell proliferation assay using flow cytometry”. However, there is no description of the flow cytometry in it, because the analysis is described in another section 4.8. The sentences starting in lines 178 and 183 are unclear.
  4. Why proliferation in cancer cell co-cultures with CIK cells assessed by CCK-8 assay is only assigned to A-498 or Caki-2 cells?
  5. It is confusing when a fixed p value is presented for the series of comparisons. It is not clear, to which groups of comparison the p values are assigned in the figure legends.
  6. Figure 7 has no A and B panels, but they are discussed in text.
  7. Why there are no controls for IFN-gamma levels measured in CIK cell monocultures treated with nivolumab and ipilimumab? They have to be presented for correct interpretation of the results.
  8. In Discussion section, the authors refer very little to the results of their own study. The data have to be better analysed and discussed.
  9. Authors conclude that CIK cells upregulate interferon-gamma (line 478), however, this is based on insignificant result and without CIK monoculture data with the immunomodulators.

Minor revision:

  1. In the captions of Fig. 5 and 6, there are statements that cells were treated “with different ratios of nivolumab” or “different ratios of ipilimumab”. What were these ratios? Why they are not indicated in the legends/bar labels?
  2. Materials and Methods section is the second in the MS, however, it is numbered 4.
  3. There are abbreviations without explanations used in text and in figure legends. For example, E:T (one might assume this is for the effector and target cells, however, it is not explained).
  4. The labels for bars with the same treatments are presented in different way on different plots. The labelling should be unified throughout the MS.
  5. The font should be unified throughout the text.
  6. In some places “+” and “-“ for CD3 and CD56 is used as superscript, but in others – not.
  7. No superscript for “4” indicating cell number in line 303. Why it was important to indicate the initial cell number?
  8. Line 241 statement about regression method for ELISA would better fit next to the ELISA assay description that to Statistical analysis.
  9. In line 412, it is indicated Figure 9, however, the results discussed belong to the Figure 10.

Author Response

Dear Dr. Reviewer 1,

thank you very much for considering our manuscript for your journal. We made the requested changes to the manuscript as described in the point by point answer below.

Major revision:

1.     What was the reason to investigate the effect of nivolumab and ipilimumab on the proliferation of CIK cells for 11 days in monoculture (Fig. 4) while the treatments of cancer cells and co-cultures lasted 72 hours? What is the difference between the results presented in Fig. 4, Fig. 5 C and Fig. 6 C? If panels C for Fig. 5 and Fig. 6 are completely the same, why they are shown in different figures?

Reply: Based on the Methods of CIK expansion in this manuscript, on day 8, cell population was heterogeneous and showed >90% CD3+, >70% CD8+ T cells, >20% CD3+CD56+ cells, and <5% CD3−CD56+ cells; this population was named CIK cells (Schmidt-Wolf IG, et al. J Exp Med. 1991. Use of a SCID Mouse/Human Lymphoma Model to Evaluate Cytokine-induced Killer Cells with Potent Cell Activity). To detect the proliferation ability, co-culture time of CIK cells and cancer cells lasted at least 48 hours. Fig.4 (now Fig. 5B) demonstrated the number of viable CIK cells after treatment of immune checkpoint inhibitors by Trypan-blue. Fig.5C (now Fig. 6B) showed us the number of viable CIK cells after treatment of immune checkpoint inhibitors in the presence of Caki-2 by Trypan blue. Additionally, Fig. 6C (now Fig. 7B) indicated the number of viable CIK cells after treatment of immune checkpoint inhibitors in the presence of A498. The figures might be not clear. We have revised the labels of these figures to make clear to readers. Thank you.       

2.     In panels A and B of Figures 5, there are the same data for the first two and the last bars. The same is in Fig. 6. It would be reasonable to combine those two figures to one and to avoid the repeated data.

Reply: We have combined panel A and B two figures to one of Figure 5 (now Fig 6A). Thank you very much for your comment.

3.     The methods section seems overloaded with details yet unclear and has to be thoroughly revised for English language. Some details brings in confusion. For example, why there is a range of dilutions used for trypan blue staining? Or what does it mean the statement “according to the manuscript” in line 160, in the description of CCK-8 assay? The section 4.7. is named “Cell proliferation assay using flow cytometry”. However, there is no description of the flow cytometry in it, because the analysis is described in another section 4.8. The sentences starting in lines 178 and 183 are unclear.

Reply: We have revised some sentences in the Methods section. Based on the original cell  concentration, the author used the different dilutions ratio of trypan blue. In the line 160 (now line 418), it means according to the protocol. We renamed the section 4.7. (now section 4.8.) “Cell proliferation assay using CellTrace™ Violet Cell Proliferation Kit”. We have revised the sentences from line 169 to line 183 (new MS line 445-455).

4.     Why proliferation in cancer cell co-cultures with CIK cells assessed by CCK-8 assay is only assigned to A-498 or Caki-2 cells?

Reply: CCK-8 assay only reflects the total cell proliferation including CIK cells and cancer cells when they are co-cultured together. Due to the limitation of this experiment, we further conduced flow cytometry to clarify the effect of PD-1/ CTLA-4 on the proliferation of CIK cells or cancer cells.

5.     It is confusing when a fixed p value is presented for the series of comparisons. It is not clear, to which groups of comparison the p values are assigned in the figure legends.

Reply: We have explained again the p values meaning in comparison to groups, especially in the figure legends.  

6.     Figure 7 has no A and B panels, but they are discussed in text.

Reply: Yes. In Results Section we replaced it with Figure 8 in the text (new line 233). 

7.     Why there are no controls for IFN-gamma levels measured in CIK cell monocultures treated with nivolumab and ipilimumab? They have to be presented for correct interpretation of the results.

Reply: We have supplementary data of nivolumab control of IgG4 isotype and ipilimumab of IgG1 isotype. 

8.     In Discussion section, the authors refer very little to the results of their own study. The data have to be better analysed and discussed.

Reply: We revised them in Discussion section.

9.     Authors conclude that CIK cells upregulate interferon-gamma (line 478), however, this is based on insignificant result and without CIK monoculture data with the immunomodulators.

Reply: Yes. Based on our new data, there was a significant elevation of interferon-gamma after a combination of nivolumab and ipilimumab in a CIK and A498 co-culture system, whereas there was no effect on Caki-2 cells.   

Minor revision:

1.     In the captions of Fig. 5 and 6, there are statements that cells were treated “with different ratios of nivolumab” or “different ratios of ipilimumab”. What were these ratios? Why they are not indicated in the legends/bar labels?

Reply: In Fig.5 and Fig.6 (now Fig.6 and Fig.7 Legends), CIK cells and A498 or Caki-2 cells were co-cultured at a ratio 10:1 with either 20ug/ml nivolumab or 20ug/ml ipilimumab treatment or a combination treatment.        

2.     Materials and Methods section is the second in the MS, however, it is numbered 4.

Reply: We rearranged them.

3.     There are abbreviations without explanations used in text and in figure legends. For example, E:T (one might assume this is for the effector and target cells, however, it is not explained).

Reply: We have explained it in Fig.3 Legend and new line 106 in the text.

4.     The labels for bars with the same treatments are presented in different way on different plots. The labelling should be unified throughout the MS.

Reply: We have changed the labels for bars accordingly.

5.     The font should be unified throughout the text.

Reply: We have revised it. Thank you very much for your comment.

6.     In some places “+” and “-“ for CD3 and CD56 is used as superscript, but in others – not.

Reply: We have revised all of them as superscript including superscript in Fig 1.

7.     No superscript for “4” indicating cell number in line 303. Why it was important to indicate the initial cell number?

Reply: We have corrected it as 5 x 104 in line 303 (new line 156). For the aim of recording increasing cell number of CIK cells for assessment of proliferation using Trypan blue, the initial cell number is important.

8.     Line 241 statement about regression method for ELISA would better fit next to the ELISA assay description that to Statistical analysis.

Reply: “ The 4PL nonlinear regression model was used for ELISA calibration curve”. We descripted it as “The data was analyzed using MARS data analysis with Four-Parameter Curve Fit (BMG Labtech, Orthenberg, Germany) ” in line 241 (new line 491) in Section 4.10. ELISA assay.  

9.     In line 412, it is indicated Figure 9, however, the results discussed belong to the Figure 10.

Reply: We have revised it in line 412 (new line 264), the results of Figure 11 were discussed in the text.

In summary we closely followed your suggestions. We hope that our manuscript is now suitable for publication in your journal.

Thank you very much.

Best regards,

Prof. Ingo Schmidt-Wolf

for the authors

Reviewer 2 Report

In this study, Naghizadeh  et al investigated the potential effects of administering nivolumab and  ipilimumab on cytotoxicity of CIK cells in culture with the RCC A-498 and Caki-2. Thy observed that PD-1 and PD-L1 blockade combination strengthened the tumoricidal activity of CIK cells on RCC. They concluded that combination of immune checkpoint inhibitors with CIK cells may represent a potential approach in treatment of patients with renal carcinoma.  The study is sound and may be great interest to readers.

It is a bit difficult to follow the reasoning when reading the manuscript. For instance, why did the authors decide to evaluate nivolumab and ipilimumab. This need to be well inserted in the introduction. In addition, the instruction needs to be a little detailed and specific with latest information to allow readers appreciate the findings.

Otherwise study is sound and may be great interest to readers.

Minor

Caki-2. Vs CAKI-2. Line 192

Author Response

Dear Dr Reviewer 2,

thank you very much for considering our manuscript for your journal. We made the requested changes to the manuscript as described in the point by point answer below.

Comments and Suggestions for Authors

In this study, Naghizadeh  et al investigated the potential effects of administering nivolumab and  ipilimumab on cytotoxicity of CIK cells in culture with the RCC A-498 and Caki-2. Thy observed that PD-1 and PD-L1 blockade combination strengthened the tumoricidal activity of CIK cells on RCC. They concluded that combination of immune checkpoint inhibitors with CIK cells may represent a potential approach in treatment of patients with renal carcinoma.  The study is sound and may be great interest to readers.

It is a bit difficult to follow the reasoning when reading the manuscript. For instance, why did the authors decide to evaluate nivolumab and ipilimumab. This need to be well inserted in the introduction. In addition, the instruction needs to be a little detailed and specific with latest information to allow readers appreciate the findings.

Otherwise study is sound and may be great interest to readers.

Reply: We have revised the Method section and Introduction section. Thank you for your comments. Additionally, we added PD-1 and CTLA-4 expression on CIK cells and PD-L1/PD-L2 expression on tumor cells in the Figure 2. in the new manuscript.         

Minor

Caki-2. Vs CAKI-2. Line 192

Reply: We have revised it in line 192 (now line 460 in new MS).

In summary we closely followed your suggestions. We hope that our manuscript is now suitable for publication in your journal.

Thank you very much.

Best regards,

Prof. Ingo Schmidt-Wolf

for the authors

Round 2

Reviewer 1 Report

In general, the study is significantly improved in presentation style. There still are some minor points that could be revised before publication.

  1. Regarding the reply to the comment 1, there still is some confusion about the results presented in Figure 5. In the Figure title it is clearly stated that the effects of nivolumab and ipilimumab on proliferation of CIK cells were tested after 11 DAYS OF MONOCULTURE. Were are the conditions of the monoculture treatment described? In text you say that the cells were treated with the antibodies for 3 days. At what day of monoculture the treatment started? At day 8? Day 11? Or the treatment with antibodies lasted 11 days? I think the procedure of monoculture treatments should be included in Methods, and also the title of the Figure 5 has to be modified to more precise version to avoid confusion.
  2. Regarding the reply to the comment 7, there are supplementary data mentioned, however, there are no supplementary data mentioned in the MS?
  3. The beginning of the sentence in line 476 has to be changed to “To investigate whether nivolumab or ipilimumab affect the proliferation of CIK cells,…”
  4. Line 483 “were” has to be changed to “was”

Author Response

Dear Editor,

thank you very much for considering our manuscript for your journal. We made the requested changes to the manuscript as described in the point by point answer below.

  1. Regarding the reply to the comment 1, there still is some confusion about the results presented in Figure 5. In the Figure title it is clearly stated that the effects of nivolumab and ipilimumab on proliferation of CIK cells were tested after 11 DAYS OF MONOCULTURE. Were are the conditions of the monoculture treatment described? In text you say that the cells were treated with the antibodies for 3 days. At what day of monoculture the treatment started? At day 8? Day 11? Or the treatment with antibodies lasted 11 days? I think the procedure of monoculture treatments should be included in Methods, and also the title of the Figure 5 has to be modified to more precise version to avoid confusion.

        Reply: Actually, we treated CIK cells with antibodies at CIK cells on day 8. After 72hs, we harvested CIK cells on day 11. We have explained this in the manuscript. Thank you.

  1. Regarding the reply to the comment 7, there are supplementary data mentioned, however, there are no supplementary data mentioned in the MS?

        Reply: We have descripted this in detail in the Discussion and the Method Section.  

  1. The beginning of the sentence in line 476 has to be changed to “To investigate whether nivolumab or ipilimumab affect the proliferation of CIK cells,…”

        Reply: OK. Thank you for your comment. We have revised in line 519 in the new version of our manuscript.

   4. Line 483 “were” has to be changed to “was”

       Reply: It has been modified in line 511 of the new version of our manuscript. Thank you.

In summary we closely followed your suggestions. We hope that our manuscript is now suitable for publication in your journal.

Thank you very much.

Best regards,

Prof. Ingo Schmidt-Wolf

for the authors

This manuscript is a resubmission of an earlier submission. The following is a list of the peer review reports and author responses from that submission.